# Fused Enzyme Glucose-6-Phosphate Dehydrogenase::6-Phosphogluconolactonase (G6PD::6PGL) as a Potential Drug Target in *Giardia lamblia*, *Trichomonas vaginalis*, and *Plasmodium falciparum*

**DOI:** 10.3390/microorganisms12010112

**Published:** 2024-01-05

**Authors:** Laura Morales-Luna, Montserrat Vázquez-Bautista, Víctor Martínez-Rosas, Miriam Abigail Rojas-Alarcón, Daniel Ortega-Cuellar, Abigail González-Valdez, Verónica Pérez de la Cruz, Roberto Arreguin-Espinosa, Eduardo Rodríguez-Bustamante, Eden Rodríguez-Flores, Beatriz Hernández-Ochoa, Saúl Gómez-Manzo

**Affiliations:** 1Laboratorio de Bioquímica Genética, Instituto Nacional de Pediatría, Secretaría de Salud, Mexico City 04530, Mexico; lauraeloisamorales@gmail.com (L.M.-L.); montsevazquez97@gmail.com (M.V.-B.); ing_vicmr@hotmail.com (V.M.-R.); mrm.roa26@gmail.com (M.A.R.-A.); 2Posgrado en Ciencias Biológicas, Universidad Nacional Autónoma de México, Mexico City 04510, Mexico; 3Programa de Posgrado en Biomedicina y Biotecnología Molecular, Escuela Nacional de Ciencias Biológicas, Instituto Politécnico Nacional, Mexico City 11340, Mexico; 4Laboratorio de Nutrición Experimental, Instituto Nacional de Pediatría, Secretaría de Salud, Mexico City 04530, Mexico; dortegadan@gmail.com; 5Departamento de Biología Molecular y Biotecnología, Instituto de Investigaciones Biomédicas, Universidad Nacional Autónoma de México, Mexico City 04510, Mexico; abigaila@biomedicas.unam.mx; 6Neurobiochemistry and Behavior Laboratory, National Institute of Neurology and Neurosurgery “Manuel Velasco Suárez”, Mexico City 14269, Mexico; veped@yahoo.com.mx; 7Departamento de Química de Biomacromoléculas, Instituto de Química, Universidad Nacional Autónoma de México, Mexico City 04510, Mexico; arrespin@unam.mx (R.A.-E.); erodriguezb@gmail.com (E.R.-B.); eden_rf@ciencias.unam.mx (E.R.-F.); 8Departamento de Bioingeniería, Escuela de Ingeniería y Ciencias, Tecnológico de Monterrey, Monterrey 64849, Mexico; 9Laboratorio de Inmunoquímica, Hospital Infantil de México Federico Gómez, Secretaría de Salud, Mexico City 06720, Mexico

**Keywords:** bifunctional enzyme, NADPH, parasitosis, enzymatic inhibition

## Abstract

Several microaerophilic parasites such as *Giardia lamblia*, *Trichomonas vaginalis*, and *Plasmodium falciparum* are major disease-causing organisms and are responsible for spreading infections worldwide. Despite significant progress made in understanding the metabolism and molecular biology of microaerophilic parasites, chemotherapeutic treatment to control it has seen limited progress. A current proposed strategy for drug discovery against parasitic diseases is the identification of essential key enzymes of metabolic pathways associated with the parasite’s survival. In these organisms, glucose-6-phosphate dehydrogenase::6-phosphogluconolactonase (G6PD:: 6PGL), the first enzyme of the pentose phosphate pathway (PPP), is essential for its metabolism. Since G6PD:: 6PGL provides substrates for nucleotides synthesis and NADPH as a source of reducing equivalents, it could be considered an anti-parasite drug target. This review analyzes the anaerobic energy metabolism of *G. lamblia*, *T. vaginalis*, and *P. falciparum*, with a focus on glucose metabolism through the pentose phosphate pathway and the significance of the fused G6PD:: 6PGL enzyme as a therapeutic target in the search for new drugs.

## 1. Parasitic Infections with an Impact on the Population

Parasitic infections are widely distributed throughout the world and are considered to be one of the primary public health concerns due to their high prevalence, especially in developing countries in Asia, Africa, and Latin America. Additionally, they are also present in developed countries, primarily due to the migration of people from third-world countries. Therefore, the World Health Organization (WHO) considers parasites to be one of the leading causes of morbidity, affecting a significant portion of the population.

Worldwide, parasitic diseases exhibit a high prevalence due to various physiological, immunological, genetic, and nutritional factors. These factors, in conjunction with the socioeconomic and cultural conditions of a population, favor the presence of these diseases [1]. Among clinically relevant protozoa, this work will focus on three parasites which cause human disease: *Giardia lamblia*, the etiological agent of giardiasis, an infection that is among the main causes of gastroenteritis associated with parasitic infections [2]; *Trichomonas vaginalis*, a parasite that colonizes the vaginal mucosa, is the etiological agent of trichomoniasis, a disease classified as one of the most common non-viral sexually transmitted infections (STIs) worldwide [3,4,5]; and the parasite *Plasmodium falciparum*, the etiological agent of human paludism or malaria [6,7] (Figure 1).

### 1.1. Giardiasis

Giardiasis is a gastrointestinal disease in humans caused by the parasite *Giardia lamblia*. The parasite has two stages in its life cycle: the cyst is the infective form, and the trophozoite is the proliferative phase responsible for clinical manifestations in the host (Figure 1) [8]. The disease may be associated with poor living conditions and inadequate hygiene, leading to a higher incidence in developing countries, with approximately 200 million cases reported worldwide each year [8,9]. Clinical manifestations of giardiasis can occur either asymptomatically or symptomatically. In asymptomatic patients, detecting the disease is challenging and increases the potential for transmission since these individuals can still spread the parasite [10,11]. Patients with symptomatic clinical disease manifest severe gastrointestinal disorders because trophozoites adhere to the host’s intestinal epithelium, disrupting the absorption of nutrients and water [10]. In severe cases, patients develop complications that lead to gastrointestinal disorders such as irritable bowel syndrome, malabsorption syndrome, and malnutrition [11,12,13]. The population with the highest incidence and severity of symptoms is children under five years old, who may experience delays in both physical and intellectual development [14]. The first-line of treatment for giardiasis is the metronidazole (MTZ) drug. In cases of MTZ intolerance, derivatives like secnidazole, nimorazol, timidazol, and albendazole, among others, are also prescribed [13,15]. However, *G. lamblia* has also shown the issue of strains developing resistance to treatments [11,12,13]. Therefore, it is essential to address the search for new treatments for this disease.

### 1.2. Trichomoniases

Trichomoniasis is a sexually transmitted disease (STD) caused by the parasite *Trichomonas vaginalis*, which belongs to the taxonomic group within the phylum Excavata. This parasite colonizes the urogenital tract in humans, causing the disease (Figure 1). Epidemiological data report approximately 270 million new infections per year, with a higher incidence among individuals in their reproductive age, typically between 15 and 49 years old [16,17,18]. In males, *T. vaginalis* primarily colonizes the bladder, seminal vesicles, urethra, and prostate, while in females, it resides in the vagina and cervix. Despite being classified as an STD, the parasite can be transmitted not only through sexual contact but also through contact with common objects such as bath towels, bath sponges, and contaminated toilets. It can even be transmitted from an infected mother to her newborn child [19,20,21]. The clinical manifestations of the disease encompass a spectrum, ranging from the production of odorous vaginal discharge, itching, and pain during urination to severe complications that can adversely affect the reproductive health of patients. These significant issues encompass the development of cervical neoplasia, cervical carcinoma, infertility and pelvic inflammatory disease, and in pregnant women, trichomoniasis can result in preterm birth or even fetal loss [22,23,24]. Regarding treatment, the MTZ drug is typically prescribed 2 g orally in a single dose. Longer treatment regimens may include 500 mg of MTZ 2 times/day for 7 days or the utilization of alternative or combination therapies involving drugs such as tinidazole and secnidazole [25]. The availability of drugs for trichomoniasis treatment is limited, and there have been reports of strains displaying resistance to conventional treatments [26]. This issue necessitates attention through the pursuit of novel therapeutic approaches to mitigate the impact of the infection on the sexual and reproductive health of individuals in their reproductive years as this has the potential to pose a challenge for future generations.

### 1.3. Malaria

Malaria, also known as malaria fever, is an acute febrile disease which is potentially fatal and is caused by parasites of the *Plasmodium* genus. It is transmitted to humans through the bite of the female *Anopheles* mosquito. There are five species of parasites that can cause malaria in humans, but *Plasmodium falciparum* and *Plasmodium vivax* are the greatest threat (Figure 1) [27]. While malaria is a preventable and curable disease, without prompt diagnosis and effective treatment, an uncomplicated case of malaria can progress to a severe form of the disease, often leading to a fatal outcome if left untreated. The WHO has classified malaria as a priority disease due to its high incidence rates, particularly in tropical regions (WHO, 2023 (Global report on neglected tropical diseases)) [28]. The African continent continues to report the highest burden of malaria globally. In 2021, the region registered 234 million cases, constituting 95% of all malaria cases, with 593,000 of them resulting in death (96% of all malaria-related deaths). Nearly 80% of these fatalities occurred in children under the age of 5 years. Remarkably, four African countries, Nigeria (26.6%), the Democratic Republic of the Congo (12.3%), Uganda (5.1%), and Mozambique (4.1%), accounted for nearly half of all malaria cases worldwide [29].

The primary symptoms of malaria include fever, headache, chills, back pain, cough, weakness, nausea, and vomiting. These symptoms can progress to severe manifestations that may lead to multi-organ failure, coma, acute renal failure, pulmonary edema, severe anemia, acidosis, hemorrhage, cerebral malaria, and, ultimately, the patient’s death [29,30,31,32]. For the treatment of malaria, various drugs are used that target different intracellular pathways in the parasite. Antimalarial drugs include quinoline derivatives, folate antagonists, and artemisinin-based combination therapy (ACT), with recommended drugs for combination therapy such as lumefantrine, sulfadoxine, pyrimethamine, and mefloquine, as advised by the WHO. However, despite malaria being a curable disease, over the past decade, the resistance of parasites to antimalarial drugs has emerged as a significant threat in the fight against malaria. Recent reports of drug-resistant malaria cases in Africa have also raised concerns with the WHO [33,34,35]. This issue underscores the critical need to discover and develop new antimalarial drugs with distinct mechanisms of action compared to currently used medications.

In recent years, efforts to develop new antimalarials have been primarily focused on conducting phenotypic studies of compound libraries to identify small molecules that inhibit the growth of *P. falciparum* in the blood-stage asexual phase. This approach has led to the identification of several million compounds and the discovery of tens of thousands of inhibitory molecules [36,37,38,39].

## 2. Treatment and Drug Resistance in Giardicidal, Trichomonicidal, and Antimalarial Therapies

### 2.1. Resistance to Nitroimidazole Drug Family

For the treatment of infections caused by *G. lamblia* and *T. vaginalis*, drugs belonging to the nitroheterocycle family are commonly employed, primarily MTZ, nitazoxanide, (NTZ), and furazolidone (FUR) [40]. In the case of *G. lamblia*, benzimidazole-derived drugs, including albendazole (ABZ) and mebendazole (MBZ), are also utilized [41,42]. Nevertheless, drug-resistant strains of *G. lamblia* and *T. vaginalis* have been isolated from patients [43]. The biological activity of nitroheterocycles compounds is in part due to the oxidation-reduction process by oxidoreductase enzymes (pyruvate ferredoxin oxidoreductases and nitroreductases) in metabolically active parasites. Upon reduction i MTZ, it induces oxidative stress, damaging the parasites’ proteins and DNA [44,45,46]. Resistance of *G. lamblia* and *T. vaginalis* strains to nitroheterocycles has been associated with the differential regulation of oxidoreductase enzymes in parasites [45,47,48,49,50]. On the other hand, the mechanism of action in *G. lamblia* for benzimidazole family compounds involves their binding to β-tubulin, preventing microtubule polymerization and disrupting microtubules. However, specific point mutations occur mainly in β-tubulin, changing specific amino acid glutamic acid to lysine at position 198 (E198K), and the benzimidazole fails to kill *G. lamblia* [12,51].

### 2.2. Resistance to Antimalarials

Among the drugs prescribed for the treatment of malaria are quinoline derivatives (quinacrine and chloroquine) and the use of artemisinin. However, *P. falciparum* has developed clinical resistance to all commonly used antimalarial drugs, including artemisinin-based combination therapy, which allows the parasite to evade the toxic effects of the drugs [52]. Resistance of *P. falciparum* to quinoline derivatives is associated with mutations in two genes encoding transporter proteins, the *P. falciparum* chloroquine resistance transporter (*Pfcrt*) and the *P. falciparum* multidrug resistance gene 1 (*Pfmdr1*) [53,54]. The *Pfmdr1* gene was established as a key factor in the development of drug resistance in *P. falciparum’s* tolerance to a variety of drugs, including quinine, chloroquine, and mefloquine [55,56], where their augmented expression of *Pfmdr* avoids intra-parasite-drug accumulation and therefore the therapeutic drug effect. On the other hand, resistance of *P. falciparum* to the antimalarial artemisinin has been associated with specific mutations in the *Kelch13* gene (*PfK13*) [57,58,59]. These mutations result in a reduction in the expression of the K13 protein, and although the precise role of the K13 protein is not clear, it has been suggested that the protein regulates the biogenesis of digestive vacuoles, as well as the uptake and degradation of hemoglobin. Consequently, reduced K13 expression or mutations in K13 lead to a decrease in heme derived from hemoglobin, resulting in resistance to artemisinin. The activation of the drug depends on heme to subsequently damage the parasite’s proteins and lead to its death [52,60,61].

## 3. Search for New Targets in Drug Design

Due to the issue of drug resistance in the parasites *G. lamblia*, *T. vaginalis*, and *P. falciparum*, as described in the previous section, there is an urgent need for the development of new, effective, and specific antiparasitic agents. The search for and development of new drugs is not a simple process; one of the strategies in drug development involves the identification of enzymes in pathogenic organisms that are crucial for their metabolism and survival. These enzymes become well-recognized targets offering a degree of specificity as they do not overlap with human homologous proteins. This strategy is known as a target-based approach [62,63,64,65]. The study of anaerobic metabolism in amitochondrial parasites like *G. lamblia* and *T. vaginalis* has yielded significant discoveries, providing potential targets for developing new and effective therapeutic alternatives for pathologies associated with these parasites. In this regard, the discovery and development of compounds that affect the energy metabolism of these parasites could lead to an improvement in treatment. By disrupting the glucose-catalyzing pathways, it could result in a reduction in the viability of the pathogen parasites.

In this context, some research groups have been conducting a search for enzyme inhibitors involved in the glycolytic pathway as a new approach in the treatment of giardiasis and trichomoniasis. An example is the inhibition of the enzyme triosephosphate isomerase in both *G. lamblia* and *T. vaginalis* [66,67]. This enzymatic inhibition leads to a disruption in the glycolytic pathway and an accumulation of toxic metabolites like methylglyoxal [68], resulting in a decrease in trophozoite viability. These findings suggest that inhibiting glucose metabolism offers an alternative for treating these parasitic infections.

## 4. Carbohydrate Metabolism and Role of the Pentose Phosphate Pathway in Parasites

One of the strategies for the discovery of new drugs involves the identification and selection of targets, the search for hit compounds that have an affinity for the target, followed by the optimization of the hit compounds, and finally preclinical and clinical studies [69]. Once the target in the pathogen of interest has been identified and characterized, compounds are needed to initiate the search for candidate drugs through high-throughput screening (HTS). This screening aims to evaluate whether the compounds possess the property of interacting with and binding to the target, thus affecting its activity. Subsequently, it can be assessed whether the selected compounds trigger enzymatic reactions and/or activate signaling pathways.

### 4.1. Glucose Metabolism

*G. lamblia* and *T. vaginalis* are amitochondrial and asexual microorganisms that lack several characteristics of higher eukaryotic respiratory metabolism, such as the presence of cytochromes and oxidative phosphorylation [70,71]. These parasites belong to the group of microorganisms classified as microaerophiles [72,73,74,75]. Their metabolism is primarily glycolytic and fermentative [31,72,76]. The final step of glycolysis is catalyzed by the anaerobic bacterial homolog enzyme pyruvate:ferredoxin oxidoreductase (PFOR), which substitutes the aerobic pyruvate dehydrogenase enzyme [31,77]. In *G. lamblia*, the fermentation end products include CO_2_, acetate, alanine, and ethanol, which vary depending on the growth conditions [31,32,78]. In *T. vaginalis*, the final products are glycerol and succinate, followed by an additional conversion of pyruvate and malate to hydrogen and acetate in an organelle called the hydrogenosome. In these parasites, the tricarboxylic acid pathway, which typically occurs in the mitochondria, is absent, although they possess the enzyme malate dehydrogenase [79].

*G. lamblia*, in addition to the enzyme PFOR, has the arginine dihydrolase pathway to provide alternative sources of energy [31,80,81,82]. The amino acid L-Arginine serves as the fuel in the early and more proliferative growth stages. The pathway consists of three enzymatic steps involving the enzymes arginine deiminase (ADI), ornithine decarboxylase (OTC), and carbamate kinase (CK) for ATP production. In this pathway, the ATP yield is only one molecule for each L-Arg used. However, *G. lamblia* can produce ATP more quickly through this pathway than from glycolysis [83]. In the case of *T. vaginalis*, the hydrogenosome contains electron transport components linked to the PFOR enzyme, a hydrogenase, and a terminal oxidase. It has also been described that hydrogenosomes store calcium [16]. In summary, the glucose oxidation of amitochondrial parasites produce pyruvate, which is converted to acetyl-CoA by the action of PFOR; subsequently, acetyl-CoA can be converted to acetate with concomitant production of ATP or acetaldehyde and then ethanol.

### 4.2. Pentose Phosphate Pathway: A Potential Origin of Therapeutic Targets

In addition to glycolysis, glucose is also metabolized through the Pentose Phosphate Pathway (PPP). The PPP is a key metabolic route, and as such, it has been proposed as a strategic pathway for target discovery in drug design to combat infectious diseases, including those caused by parasites such as *G. lamblia*, *T. vaginalis*, and parasites of the *Plasmodium* genus. The PPP comprises both an oxidative phase, which is important for maintaining cellular redox homeostasis, and a non-oxidative phase, where precursors for nucleic acids and aromatic amino acids are generated.

In mammals, like humans, the oxidative phase of the PPP is considered unidirectional. Glucose-6-phosphate (G6P) undergoes metabolism through three enzymatic reactions (Figure 2). The first reaction involves the oxidation of G6P to 6-phosphogluconolactone (6PG) and the production of one NADPH molecule catalyzed by the enzyme glucose-6-phosphate dehydrogenase (G6PD). The second reaction is the hydrolysis of 6-phosphogluconolactone to 6-phosphogluconate (6PGD), catalyzed by the enzyme 6-phosphogluconolactonase (6PGL) [76,84]. During the third reaction of the oxidative phase, oxidative decarboxylation of 6-phosphogluconate occurs, catalyzed by the enzyme 6-phosphogluconate dehydrogenase (6PGDH), resulting in the production of one molecule of ribulose-5-phosphate and a second molecule of NADPH (Figure 2) [85,86,87].

Regarding the non-oxidative phase of the PPP, it consists of reversible enzymatic reactions involving enzymes such as transketolases (TKT), transaldolases (TALDO), isomerases, and epimerases (Figure 2). This phase begins with ribulose-5-phosphate, followed by a series of molecular rearrangements among various monosaccharides. The transfer of fragments containing two to three carbon atoms between monosaccharides takes place, resulting in the generation of different intermediate metabolites. Among these, glyceradehyde-3-phosphate and fructose-6-phosphate are prominent, and their fate can be directed towards glycolysis depending on the cellular requirements [86,88].

The PPP has gained importance among researchers as a target for inhibiting enzymes, mainly those in the oxidative phase, as this is where reduced NADPH molecules are generated through two reactions: one involving G6PD and the other involving 6PGDH. The NADPH produced in the oxidative phase of the PPP is essential in cells for fatty acid synthesis and for maintaining redox homeostasis as it facilitates the elimination of reactive oxygen species through the thioredoxin/peroxiredoxin systems, thereby preserving cellular homeostasis [86,89,90,91]. It is therefore reasonable to hypothesize that if an enzyme in the oxidative branch is inhibited, the NADPH concentration would decrease, leading to alterations in the redox metabolism and, in consequence, compromising stability for the parasites. This is observed in the case of *P. falciparum*, where the main source of NADPH is obtained through the PPP, even though enzymes such as glutamate dehydrogenase and isocitrate dehydrogenase (IDH) also produce NADPH. The contribution of IDH is unclear, and it has been demonstrated that glutamate dehydrogenase is dispensable for asexual growth [92]. In a study where the gene that encodes glucose-6-phosphate dehydrogenase–6-phosphogluconolactonase enzyme in *P. falciparum* was silenced using RNA interference, the result was a halt in the parasite’s growth during the trophozoite stage and increased transcription of thioredoxin reductase [93]. The results confirm that both the gene and the G6PD protein are essential for the parasite’s survival in the blood stage, suggesting a central role for the fused PfGluPho enzyme in counteracting oxidative stress [93]. In the case of 6PGDH, which catalyzes the third step of the PPP, its deletion also leads to the accumulation of 6-phosphogluconate, which inhibits phosphoglucose isomerase and, consequently, glycolysis. Moreover, the subsequent depletion of R5P production can also have an influence [94]. Therefore, enzymes of the PPP, especially those in the oxidative phase, could be promising candidates for study as new pharmacological targets aimed at improving diseases caused by eukaryotic protists like *G. lamblia*, *T. vaginalis*, and *P. falciparum*.

## 5. Structural and Kinetic Characteristic of the Bifunctional G6PD-6PGL Enzyme in Parasites

As previously we mentioned, in mammals, the first two enzymes that catalyze the oxidative phase of PPP, G6PD, and 6PGL are two separate enzymes. In *P. falciparum*, *G. lamblia*, and *T. vaginalis*, however, G6PD-6PGL is a bifunctional enzyme with both G6PD and 6PGL active sites on a single polypeptide chain of the enzyme that consists of 911, 742, and 716 amino acids, with a molecular mass of 107, 83, and 81.6 kDa, respectively, known as PfGluPho, GlG6PD::6PGL, and TvG6PD::6PGL (glucose-6-phosphate dehydrogenase::6-phosphogluconolactonase), catalyzing the first and second reactions of the PPP [84,95,96,97] (Figure 3). In this sense, Alencar et al. [98] built the first 3D structural model of the G6PD domain of the PfG6PD-6PGL enzyme. Additionally, they performed a multiple alignment of the G6PD sequences that revealed identities in the range of 33–50% with respect to G6PD enzymes from humans: *Mycobacterium avium*, *Trypanosoma cruzi*, and *Leuconostoc mesenteroides*. Furthermore, comparative analysis of the fused G6PD::6PGL enzyme from *G. lamblia* with the crystallographic structure of human G6PD (HsG6PD; PDB entry 2BH9) showed a 35% a sequence identity. Similar results were found with the TvG6PD::6PGL amino acid sequence, where the best similarity score of 37.5% was obtained with HsG6PD (PDB entry 2BH9) [96]. This characteristic has garnered attention for its potential as a target in parasites since it exhibits structural differences from the human enzyme. Because of this, these fused enzymes have been extensively characterized and proposed as potential therapeutic targets in the search for and the development of new drugs.

The fusion of the first two enzymes in the PPP (G6PD and 6PGL) was first described in two species of the Plasmodium parasite. In 1981, the first evidence of G6PD enzyme activity in *P. falciparum* (PfG6PD) was established [99]. Subsequent efforts were initiated to demonstrate both the presence of the enzyme in the parasite [100] and to characterize it. In 1990, Buran Kurdi-Haidar and Lucio Luzzatto [101] partially purified the PfG6PD enzyme from parasite extracts for the first time and described its biochemical properties. They reported a K_m_ value for G6P and NADP^+^ of 27 mM and 4.5 mM, respectively, and a Ki value for NADPH of 4.5 mM [101]. Later, the PfG6PD gene was cloned and sequenced [97,102], revealing the gene codes for a 910-amino acid protein, which was nearly twice the size of any other reported G6PD protein. It was documented that the C-terminal portion (amino acids 310-910) of the PfG6PD protein shares 39% homology with human G6PD, whereas the N-terminal portion of 300 amino acids lacks homology with human G6PD. In 2003, Clarke and colleagues [95] demonstrated that the first two steps of the PPP in *P. falciparum* and *P. berghei* are catalyzed by a novel single bifunctional enzyme named PfGluPho (glucose-6-phosphate dehydrogenase-6-phosphogluconolactonase) [95,103] (Figure 4). Additionally, these same authors reported that genes encoding very similar proteins are also found in three other malarial parasites: *Plasmodium yoelii*, *Plasmodium chabaudi,* and *Plasmodium knowlesi* [103].

A decade later, Jortzik and colleagues successfully obtained the recombinant PfGluPho fused enzyme of *P. falciparum* (PfGluPho). They determined the activity of the G6PD domain of the fused PfGluPho enzyme and the activity of the individually cloned G6PD domain. The apparent K_m_ of PfGluPho for G6P and NADP^+^ was 19.2 and 6.5 mM, respectively (Table 1), while the individually cloned G6PD domain showed a Km value of 33.2 and 6.1 mM for G6P and NADP^+^, respectively [104]. Based on these results, the authors noted that PfGluPho exhibits higher affinity for both NADP^+^ and G6P compared to HsG6PD. This implies that in infected red blood cells, where the concentrations of G6P and particularly NADP^+^ are below the K_m_ values of HsG6PD, PfGluPho utilizes most of the available G6P and NADP^+^ for NADPH generation, suggesting the significance of the PfGluPho enzyme in the parasite’s infectious stage. Given this background, the PfGluPho enzyme is proposed as a potential target for the search for new antimalarial drugs [104].

Subsequently, Stover et al. [106] conducted a comprehensive genomic analysis of various organisms in order to elucidate the origin of the gene fusion event involving *g6pd* and *6pg*l genes. Their findings not only led to the conclusion that the gene fusion event appeared to have occurred independently, near the base of metazoan and apicomplexan lineages, but also unveiled two additional fusion events. One of these events involved the fusion of the *g6pd* and *6pgl* genes in two eukaryotic parasites, *T. vaginalis* and *G. lamblia*, while the other event entailed the fusion of the *g6pd* gene with the gene responsible for encoding the third enzyme of the PPP, phosphogluconate dehydrogenase (*6pgdh*), in a diatom species [106]. However, it was observed that in the case of *T. vaginalis* and *G. lamblia*, the G6PD domain was situated in the N-terminal portion, with the 6PGL domain in the C-terminal domain (Figure 3), in contrast to Plasmodium species, *Theileria parva*, *Theileria annulate*, *Babesia bovis*, *Neospora caninum*, and *Toxoplasma gondii*, where the N-terminal domain corresponded to 6PGL and the C-terminal domain to the G6PD enzyme (Figure 3). This study broadened our understanding of the spectrum of species and medically relevant organisms featuring fusions within the PPP [106].

Afterwards, our group reported the cloning, expression, and purification of the recombinant fused protein glucose-6-phosphate dehydrogenase:6-phosphogluconate from *G. lamblia* (GlG6PD::6PGL). We also conducted a thorough characterization of the G6PD domain. The K_m_ value for the G6PD domain of the fused enzyme was found to be 18 µM for G6P and 13 µM for NADP^+^. Additionally, we reported a k_cat_ value of 31.8 s^−1^ [96]. This k_cat_ value is lower compared to the human G6PD (233 s^−1^) [105], and it is five-fold higher than the G6PD domain of the PfGluPho protein. Also, the kinetic parameters for the 6PGL domain in the fused enzyme GlG6PD::6PGL and the kinetic parameters for the individually cloned G6PD domain were provided (Table 1) [87]. The K_m_ value for the individually cloned G6PD domain was 94.2 µM for G6P and 26.7 µM for NADP^+^, both of which were higher compared to the G6PD domain in the fused enzyme GlG6PD::6PGL. This indicates a lower substrate affinity in the individual G6PD protein [87]. The K_m_ value of the 6PGL domain in the fused enzyme GlG6PD::6PGL was determined to be 51 µM for the substrate 6-phosphoglucono δ-lactone, with a k_cat_ of 31.8 s^−1^. These values are lower than those reported for the human 6PGL (K_m_ = 242 µM) and *P. falciparum* 6PGL (K_m_ = 172 µM). Unfortunately, the individually cloned 6PGL domain has not been characterized due to the inability to successfully purify it. This suggests the importance of the fusion of both genes and, consequently, the translation of a functional fused protein [87].

A possible kinetic mechanism for the GlG6PD::6PGL enzyme has also been proposed. Through mass spectrometry studies and enzymatic assays, it was demonstrated that the reaction product of the G6PD domain is released into the reaction medium. Subsequently, the 6PGL domain uptakes the product from the medium and catalyzes the production of 6-phosphogluconate (Figure 5) [87]. This finding suggests that the mechanism followed by the enzyme in organisms sharing the fusion is similar, marking a significant advancement in the understanding of the G6PD::6PGL enzyme reaction. However, no kinetic mechanisms have been proposed so far for the G6PD::6PGL fused enzymes of *T. vaginalis* and *P. falciparum*.

In the case of the fused enzyme G6PD::6PGL from *T. vaginalis*, it was subjected to kinetic characterization by Morales-Luna et al. [84]. The steady-state kinetic parameters were obtained, along with K_m_ values of 210 µM for G6P and 27 µM for NADP^+^, respectively. Notably, these values are higher than those reported for *G. lamblia*, *P. falciparum*, and *H. sapiens*. The catalytic constant (k_cat_) of TvG6PD::6PGL displayed a higher value of 147 s^−1^ compared to the parasitic counterparts, *G. lamblia* and *P. falciparum* (31 and 8 s^−1^, respectively), though it was lower than that of human G6PD (233 s^−1^) (Table 1).

From these before-mentioned studies, it becomes evident that the G6PD domain exhibits a heightened affinity for the substrate G6P when in a fused configuration with the 6PGL domain, as opposed to when they are individually cloned. This is evident as the K_m_ value for individual G6PD domains increased, indicative of diminished substrate affinity. Furthermore, the catalytic constants (k_cat_) for individually cloned domains exhibited adverse effects (Table 1). As for the individual 6PGL domains, they have not been successfully obtained through recombinant methods for any parasitic species [87,104]. This observation strongly implies that the G6PD domain imparts structural stability to the 6PGL domain within the fused enzyme, facilitating its catalytic function [81]. In this sense, Morales-Luna et al. [87] performed Molecular Dynamics (MD) Simulation of the fused G6PD::6PGL enzyme in order to explore the stability of the structure and found that when the cofactor NADP^+^ and the structural NADP^+^ were added, the fluctuations decreased, indicating that NADP^+^ had a stabilizing effect; especially, the stabilization of the 6PGL domain was more evident [84].

It has been proposed that the fusion of G6PD and 6PGL enzymes to create a bifunctional enzyme may enhance the efficiency of the oxidative phase in producing NADPH from G6P when compared to the two reactions catalyzed by individual enzymes. This arises from the immediate availability of the G6PD domain’s reaction product (6-phosphoglucono-δ-lactone) for hydrolysis by the 6PGL domain, thereby increasing the pathway’s efficiency [106]. Moreover, the presence of this fusion in *P. falciparum*, *T. vaginalis*, and *G. lamblia* could bear significant relevance during the infective phase of these parasites’ life cycles. Additionally, due to its distinct structural composition in comparison to human G6PD, it is postulated as a potential target for drug or vaccine development. Given these insights and the characterization of the fused enzymes PfGluPho, GlG6PD::6PGL, and TvG6PD::6PGL, endeavors have commenced in the quest for inhibitory compounds targeting these enzymes for potential antiparasitic drug development.

## 6. G6PD::6PGL Fused Enzyme in Parasites: Potential Drug Target

One of the strategies for the search for new drug molecules is through the evaluation of compound libraries, which include drugs with previously reported biological activity, analogs of active pharmaceutical ingredients of existing drugs, or de novo synthesized molecules. In this way, enzyme inhibitors can be discovered, and since these enzymes are vital for the target pathogen, their inhibition would lead to its demise. In this regard, various studies have been conducted to assess the inhibitory effect of chemical libraries on the fused G6PD::6PGL enzyme of the parasites *P. falciparum*, *G. lamblia*, and *T. vaginalis* [37,107,108,109,110,111,112,113,114,115]. In these studies, molecules that inhibit the enzymatic activity of the G6PD domain in the recombinant G6PD::6PGL enzyme have been identified.

Guiguemde et al. [37], employed a phenotypic forward chemical genetic approach to assay 309,474 chemicals. They disclosed in detail the profiling of 172 representative candidates and by a reverse chemical genetic study identified 19 new inhibitors of four validated drug targets and 15 novel binders among 61 malarial proteins. They identified the C276-1187 compound that has antimalarial activity and is able to bind to 6PGLPfGluPho. Afterwards, Jortzik et al. [104] studied the inhibitory activity of C276-1187 on PfGluPho, and C276-1187 showed an inhibitory effect on the 6PGL activity of PfGluPho or Hs6PGL at concentrations up to 300 μM. Furthermore, an inhibitory effect on the G6PD activity of PfGluPho was reported with an IC_50_ of 127 μM. In addition, the compound C276-1187 acts on PfGluPho’s G6PD activity as a noncompetitive-type inhibitor of both G6P and NADP^+^ substrates.

In 2012, Preuss et al. [107] reported the first discovery of small-molecule inhibitors of the recombinant enzyme PfGluPho through a target-based high-throughput screening assay. They identified 899 compounds that inhibited the activity of the G6PD domain of the PfGluPho enzyme by at least 50%. Furthermore, through structure–activity relationship (SAR) analysis, the authors reported two classes of PfGluPho inhibitors. The first class included compounds with a “pyrimidine triones” structure, such as CB22, CB61, CB62, and CB64, with IC_50_ values ranging from 4.6 µM to 19.9 µM. The second class consisted of compounds with a “chromen-2-ones” structure, and CB103 was the most potent PfGluPho inhibitor with antiplasmodial activity, featuring an IC_50_ value of 1.1 µM (Table 2). Despite not finding selective inhibitors for the PfGluPho enzyme, these PfGluPho inhibitors exhibited antimalarial activity in the micromolar range, making this study a starting point for antimalarial drug design [107].

In a subsequent study, two compounds with the property of selectively inhibiting the PfGluPho enzyme were identified, featuring IC_50_ values in the nanomolar range. One of the compounds was ((R,Z)-*N*-((1-ethylpyrrolidin-2-yl)methyl)-2-(2-fluorobenzylidene)-3-oxo-3,4-dihydro-2H-benzo[b][1,4]thiazine-6-carboxamide), referred to as ML276 (IC_50_ = 889 nM), which also exhibited in vitro activity against *P. falciparum* (IC_50_ = 2.6 µM) [108]. The second compound, (R)-*N*-((1-ethylpyrrolidin-2-yl)methyl)-4-methyl-11-oxo-10,11-dihydrodibenzo[b,f][1,4]thiazepine-8-carboxamide, known as ML304 (IC_50_ = 190 nM), was structurally related to ML276 [111]. With the results obtained with compound ML276, further molecule optimization was carried out to enhance its biological activity, leading to the development of compound SBI-0797750, which was approximately 25 times more potent than ML276 (IC_50_ = 889 nM). These findings position SBI-0797750 as a promising molecule for further studies to propose it as a potential antimalarial treatment [112].

Interestingly, a search for naturally occurring bioactive compounds to facilitate the development of novel treatments has been conducted. For example, polyphenol ellagic acid is a metabolite that is one of the active components in extracts of West African plants that are used to treat malaria in traditional medicine approaches. Allen et al. [110] identified ellagic acid as a potent inhibitor of PfGluPho, with an IC_50_ of 76 nM in the fused enzyme and IC_50_ values between 105 and 300 nM in growth inhibition, demonstrating mixed-type inhibition for both G6P and NADP^+^ substrates [110]. Additionally, ellagic acid acts synergistically with antimalarial drugs such as artesunate, chloroquine, and mefloquine [116].

As previously mentioned, Alencar et al. [98] reported the first model for the bifunctional enzyme glucose-6-phosphate dehydrogenase-6-phosphogluconolactonase (PfG6PD-6PGL) and to validate the 3D structural model of PfG6PD they performed a computational study that combined guided docking and refinement by MD simulations in order to find a stable binding mode for ML276 in the substrate-binding pocket of PfG6PD. Based on their studies, the authors proposed a binding mode of interaction with the carboxylate group of Asp750, which formed stable hydrogen bond contacts with the protonated amine and the amide NH groups of ML276. In addition, they observed that the fluorobenzene ring was firmly bound in a hydrophobic pocket shaped by Val585 and Leu778, suggesting that this binding mode would impede the binding of the substrate due to the severe steric clash between the fused ring of ML276 and the phosphate group of G6P [98]. These results were in agreement with those previously reported, where both ML276 and ML304 are the only compounds that have been found to inhibit PfG6PD through a competitive mechanism with the substrate. Based on the above, these authors synthesized a series of glucose derivatives featuring an α-methoxy group at the anomeric position and different side chains at position 6 bearing distinct basic functionalities. They reported that several compounds displayed micromolar affinity (*K*_i_ up to 23 mM), favorable selectivity (up to > 26-fold), and low cytotoxicity. However, assays with *P. falciparum* cultures revealed high micromolar IC_50_ values, likely as a result of poor internalization of the compounds in the parasite cell [98].

On the other hand, Haeussler et al. [109] cloned, produced, purified, and characterized via enzyme kinetics and inhibitor studies the *Plasmodium vivax* glucose 6-phosphate dehydrogenase (*Pv*G6PD). In addition, nanomolar IC_50_ values of several compounds such as ellagic acid, flavellagic acid, and coruleoellagic acid were determined on *P. vivax* G6PD, which are inhibitors that had been previously characterized on *Pf*GluPho. Regarding ellagic acid, it also inhibited the *P. vivax* enzyme *Pv*G6PD with an IC_50_ value of 32.5 nM, and the mechanistic characterization indicated that ellagic acid acts as a mixed-type inhibitor against both G6P and NADP^+^ in *Pv*G6PD, which is identical to the mechanism determined for the *P. falciparum* enzyme [98]. In this sense, ML304, a recently developed *Pf*GluPho inhibitor, was verified to also be active on *Pv*G6PD. The IC_50_ was found to be 190 nM on the *P. falciparum* enzyme and 80 μM on human G6PD [111]. Additionally, the mechanistic characterization was confirmed by mode of inhibition studies, which showed that ML304 acts as a competitive-type inhibitor towards G6P (*K*_i_ = 0.7 ± 0.3 μM) and as a mixed-type inhibitor towards NADP+ (*K*_i_ = 16.3 ± 8.8 μM). In addition, they identified a set of novel inhibitors of *Plasmodium* G6PD based on structure–activity relationships of the PASS system (Prediction of Activity Spectra for Substances). Finally, a new series of novel small molecules with the potential to inhibit the *falciparum* and *vivax* enzymes were synthesized, resulting in two compounds with nanomolar activity, where the 21 (vz0527) and 4 (vz1732) were the most potent compounds, with the IC_50_ reaching the nanomolar range (0.2 μM). The IC_50_ values of the two compounds on hG6PD were one to two orders of magnitude higher (8.3 μM), indicating that selective targeting of the plasmodial enzymes is possible [98].

Additionally, Berneburg et al. [112] reported an optimized dihydrodibenzothiazepine lead compound (SBI-0797750) as a potent and selective inhibitor of *Plasmodium falciparum* and *P. vivax* glucose 6-phosphate dehydrogenase 6-phosphogluconolactonase enzymes. SBI-0797750 was a potent and fully selective PfGluPho inhibitor with robust nanomolar activity against recombinant PfGluPho, PvG6PD, and *P. falciparum* blood-stage parasites. The IC50 of SBI-0797750 determined for PfGluPho and for PvG6PD was 6.7 nM and 31.0 nM, respectively. In addition, the human homologue HsG6PD was tested and an IC50 up to 99,000 nM was determined [109], indicating the high selectivity of SBI-0797750 for the plasmodial enzymes. Finally, the mode-of-action was confirmed, indicating that SBI-0797750 disturbs the cytosolic glutathione-dependent redox potential, as well as the cytosolic and mitochondrial H_2_O_2_ homeostasis of *P. falciparum* blood stages, at low nanomolar concentrations, and does not harm red blood cell (RBC) integrity and phagocytosis and, thus, does not promote anemia. SBI-0797750 is therefore a very promising antimalarial lead compound.

Regarding the bifunctional enzyme in *G. lamblia*, three studies have been conducted, in which inhibitory compounds targeting the GlG6PD::6PGL enzyme were reported. In addition to determining the IC_50_ value for the enzyme, the second-order inactivation constant *k*_2_ (µM^−1^ s^−1^) was calculated to determine the inactivation rate of each compound and to represent the rate of formation of the enzyme-inhibitor complex. In a study conducted in [114], a library of 120 compounds was assessed through inhibition assays, leading to the identification of four compounds named CNZ-7, CNZ-8, CMC-1, and FLP-2 as potent inhibitors of the GlG6PD::6PGL enzyme, with *k*_2_ constants of 0.97, 0.60, 1.88, and 0.82 M^−1^ s^−1^, respectively. These compounds induced alterations in the secondary and tertiary structure of the protein, resulting in global stability modifications and, consequently, loss of enzymatic activity. Furthermore, these compounds were tested in *G. lamblia* cultures, revealing IC_50_ values of 8.7, 15.2, 15.3, and 24.1 µM for CNZ-7, CNZ-8, CMC-1, and FLP-2, respectively [114]. Interestingly, CNZ-7, CNZ-8, and FLP-2 share structural similarities with the drug nitazoxanida since they contain the nitrothiazole ring in their structure. This is noteworthy as the antiprotozoal, antiproliferative, and anti-infectious activities of nitazoxanida have been attributed to the nitrothiazole moiety [116,117].

In a recent study by Martínez-Rosas et al. [115], it was discovered that the drug nitazoxanida acts as a catalytic inhibitor of the fused GlG6PD::6PGL enzyme in *T. vaginalis*, with a *k*_2_ value of 1.78 M^−1^ s^−1^. Consequently, the researchers proposed GlG6PD::6PGL as a new target for the drug nitazoxanida. Additionally, through fluorescence assays, it was observed that the enzyme experiences stability alterations, suggesting a loss of catalytic activity due to conformational and folding changes. The findings from this study provided new insights into the mechanism of action and a strategy for the antigiardial activity associated with nitazoxanida.

In another study by Hernández-Ochoa et al. [113], potential inhibitors for the bifunctional enzymes G6PD::6PGL from both *G. lamblia* and *T. vaginalis* were identified. In this work, five benzimidazole-type compounds, named H-BZM1, H-BZM2, H_2_N-BZM6, O_2_N-BZM7, and O_2_N-BZM9, were examined. Three of these compounds (H-BZM2, O_2_N-BZM7, and O_2_N-BZM9) had previously been reported as compounds with antigiardial activity [118]. Subsequently, the GlG6PD::6PGL and TvG6PD::6PGL enzymes were investigated as pharmacological targets for these compounds. Through inhibition assays, spectroscopic studies, and in silico computational predictions, the effects of the compounds were assessed. Two potent compounds (O2N-BZM7 and O2N-BZM9) were identified, successfully inhibiting both enzymes. In the case of GlG6PD::6PGL the *k*_2_ values were 1.92 and 6.13 M^−1^ s^−1^ for O_2_N-BZM7 and O_2_NBZM9, respectively, while for TvG6PD::6PGL, the *k*_2_ values were 0.86 and 1.62 M^−1^ s^−1^, respectively, for O_2_N-BZM7 and O_2_NBZM9 (Table 2). Additionally, when the parasites were cultured with these compounds, the IC_50_ values were determined as 14 and 17 μM for O_2_N-BZM7 and O_2_N-BZM9, respectively, in *G. lamblia*; and 6 and 4 μM for *T. vaginalis* [113].

Finally, a study conducted by Martínez-Rosas et al. [115], investigated the impact of a small library of 55 compounds on the activity of the fused *T. vaginalis* G6PD::6PGL enzyme, in which four compounds showed they could inhibit the enzyme. Three of these compounds are derivatives of nitazoxanida (CNZ-3, CNZ-17, and MCC-7), while the fourth compound is of the carboxylic acid type (JMM-3). The compounds displayed *k*_2_ values of 0.33, 0.66, 0.38, and 0.26 M^−1^ s^−1^ for JMM-3, CNZ-3, CNZ-17, and MCC-7, respectively (Table 2). CNZ-3 and MCC-7 exhibited non-competitive-type inhibition for the G6P substrate, whereas CNZ-17 and JMM-3 demonstrated uncompetitive-type inhibition for G6P. Furthermore, for the NADP^+^ substrate, CNZ-3 showed competitive inhibition, while CNZ-17, MCC-7, and JMM-3 revealed uncompetitive-type inhibition [115].

As we have described, G6PD::6PGL represents a potential and specific target that can be inhibited to counteract the effects of *G. lamblia*, *T. vaginalis*, and *P. falciparum*. In this context, we have provided information on various recently researched molecules focused on this paradigm (Figure 6). Since it is hypothesized that compounds that inhibit enzymes which participate in metabolic pathways that catalyze glucose, such as the pentose phosphate pathway, could result in a reduction in NADPH levels and consequently an alteration in the redox system of the parasites—leading to an accumulation of reactive oxygen species, in the same way the generation of nucleotide precursors, such as ribose-5-phosphate, could be reduced—so the parasites would no longer have enough nucleic acids to replicate, all these events could be generating the death of the parasites (Figure 6).

## 7. Conclusions

The limited catalog of drugs available for the treatment of parasitic diseases such as malaria, giardiasis, and trichomoniasis, along with the emergence of drug-resistant strains, presents a growing health problem. This issue could potentially lead to a lack of effective treatments for these parasitic diseases that are currently treatable. Therefore, the search for new therapies with higher efficacy, specificity, and fewer patient side effects is essential. Rational drug design is a valuable tool for identifying and evaluating specific targets that enable the development of molecules capable of inhibiting vital enzymes in clinically relevant pathogens, thereby hindering their proliferation. In this context, the fused enzyme G6PD::6PGL has been proposed as a promising specific target for the search and development of new treatments for parasitic diseases, including malaria, trichomoniasis, and giardiasis. Since this enzyme is fused exclusively in parasites, its structure differs from that of humans. Moreover, being a bifunctional enzyme, it serves as an attractive target due to its increased relevance in the cell. Therefore, several studies have focused on discovering molecules that inhibit the activity of the G6PD::6PGL enzyme and inhibit parasite growth in cultures. This paves the way for continued research in pursuit of specific treatments for infections caused by parasites harboring the bifunctional enzyme G6PD:6PGL. While further research is needed to achieve this goal, it is evident that ongoing investigations are well on track to reach this objective.

## Figures and Tables

**Figure 1 microorganisms-12-00112-f001:**
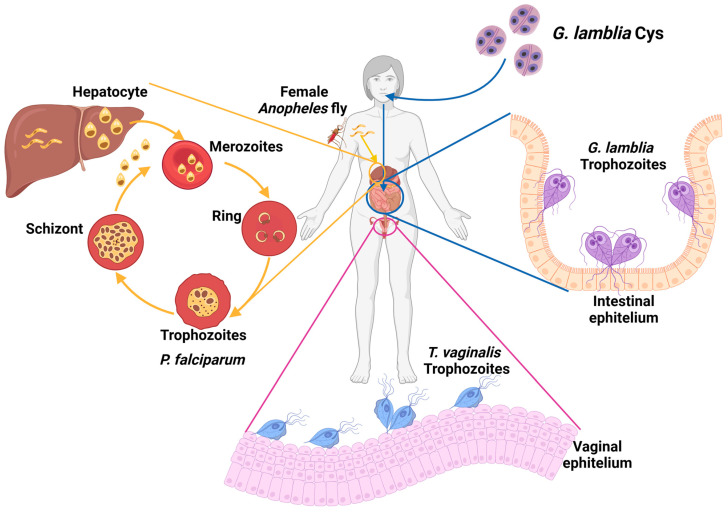
Relevant parasites that infect humans. Extracellular parasites colonize various microenvironments: *G. lamblia* colonizes the intestinal epithelium in the small intestine, and *T. vaginalis* colonizes the host’s reproductive system. On the other hand, the genus *Plasmodium* comprises species that parasitize the interior of erythrocytes within the host’s circulatory system.

**Figure 2 microorganisms-12-00112-f002:**
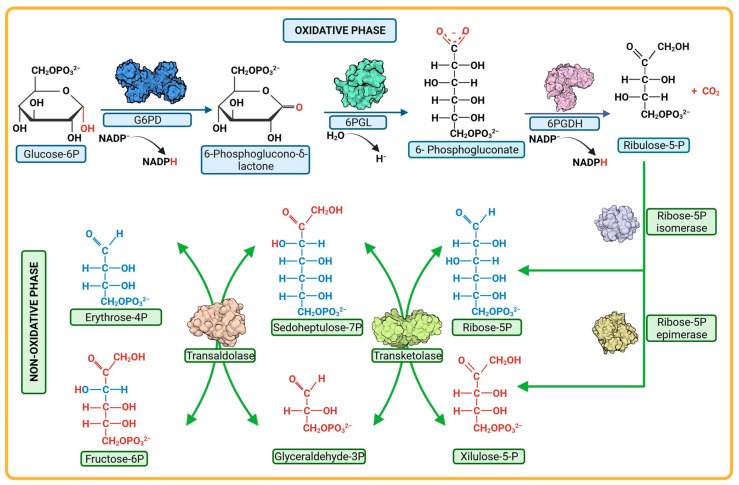
Pentose phosphate pathway. The pathway is composed of two phases, oxidative and non-oxidative. Each one comprises a series of reactions mediated by different enzymes; the enzymes G6PD, 6PGL, and 6PGDH participate in the oxidative phase. In the non-oxidative phase, the reactions are catalyzed by the enzymes epimerases, isomerase, transketolases, transaldolases.

**Figure 3 microorganisms-12-00112-f003:**
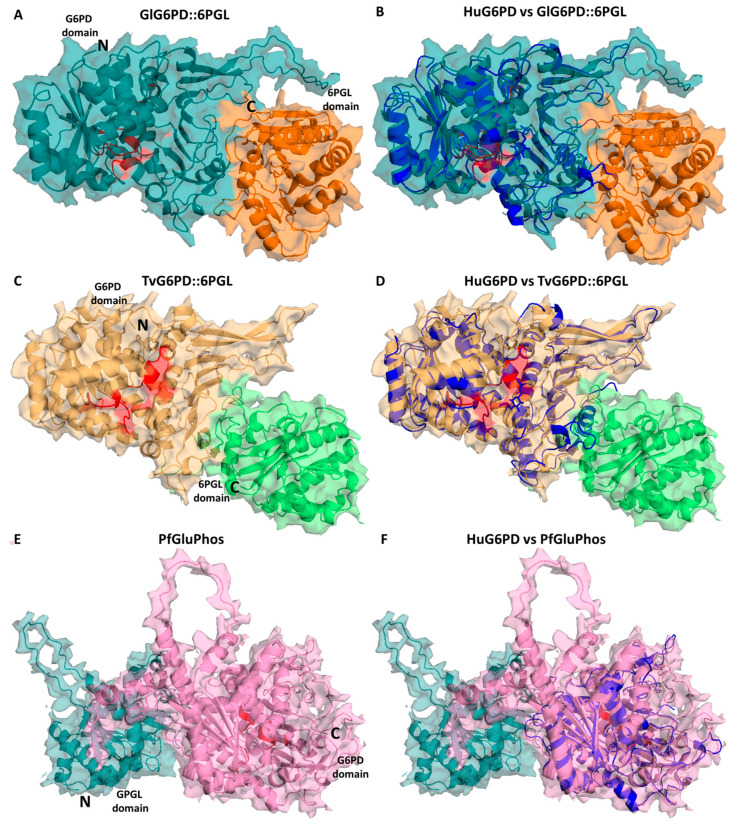
Structural alignment of the minimized G6PD::6PGL models. Structural alignment of the human G6PD enzyme (PDB entry 2BH9; blue cartoon) and the minimized model of (**A**) GlG6PD::6PGL from *G*. *lamblia*. G6PD domain is shown in blue cartoon; 6PGL domain in orange cartoon (**B**) Alignment of HuG6PD vs. GlG6PD::6PGL. (**C**) TvG6PD::6PGL from *T. vaginalis*. G6PD domain is shown in yellow cartoon; 6PGL domain in green cartoon (**D**) Alignment of HuG6PD vs. TvG6PD::6PGL. (**E**) PfGluPhos from *P. falciparum*. G6PD domain is shown in pink cartoon; 6PGL domain in cyan cartoon (**F**) Alignment of HuG6PD vs. PfGluPhos.

**Figure 4 microorganisms-12-00112-f004:**
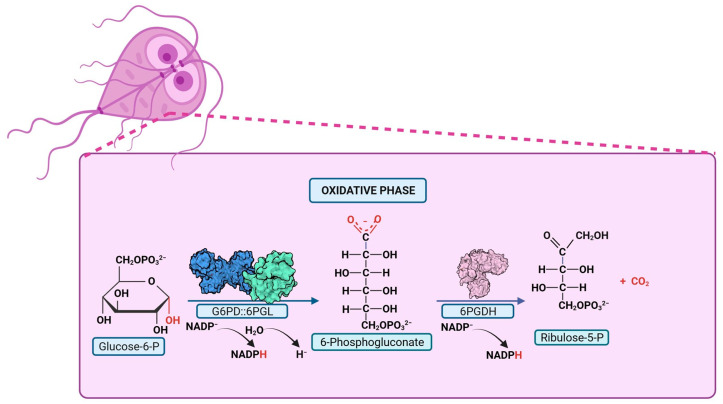
Pentose Phosphate Pathway in parasites with G6PD::6PGL fusion. PPP reactions are depicted, with the product of the bifunctional enzyme, glucose-6-phosphate dehydrogenase-6-phosphogluconolactonase, highlighted in red. The G6PD::6PGL bifunctional enzyme catalyzes the production of 6-phosphogluconate in oxidative phase.

**Figure 5 microorganisms-12-00112-f005:**
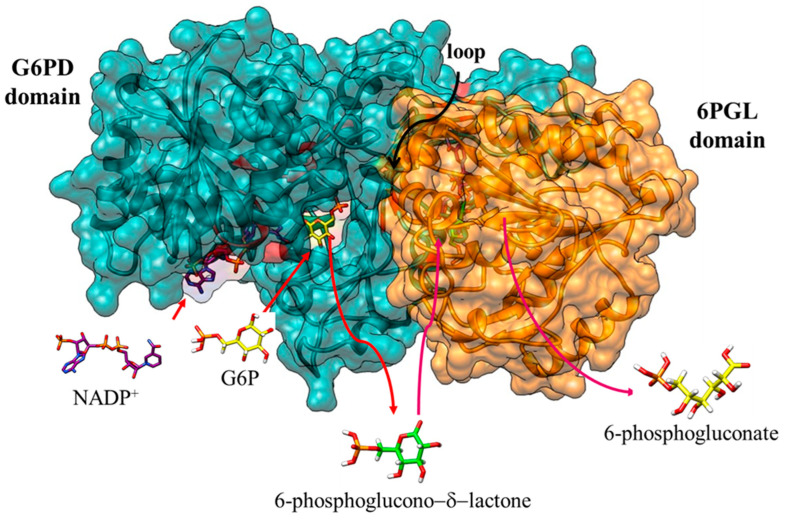
Catalysis by the fused enzyme G6PD::6PGL of *G. lamblia*. The G6PD domain model is depicted in cyan, and the 6PGL domain in orange. Substrates are represented as model structures with bars: the cofactor NADP^+^ (navy blue color) and the substrates G6P (yellow color) and 6-phosphoglucono-d-lactone (green color). Red arrows indicate the entry of G6P and NADP^+^ substrates into the active site of the G6PD domain. Red arrows signify the release of the product 6-phosphoglucono-δ-lactone into the reaction medium and its entry into the active site of the 6PGL domain. Finally, the release of 6-phosphogluconate into the reaction medium is indicated with an arrow in magenta color. With permission from Morales-Luna et al. [87].

**Figure 6 microorganisms-12-00112-f006:**
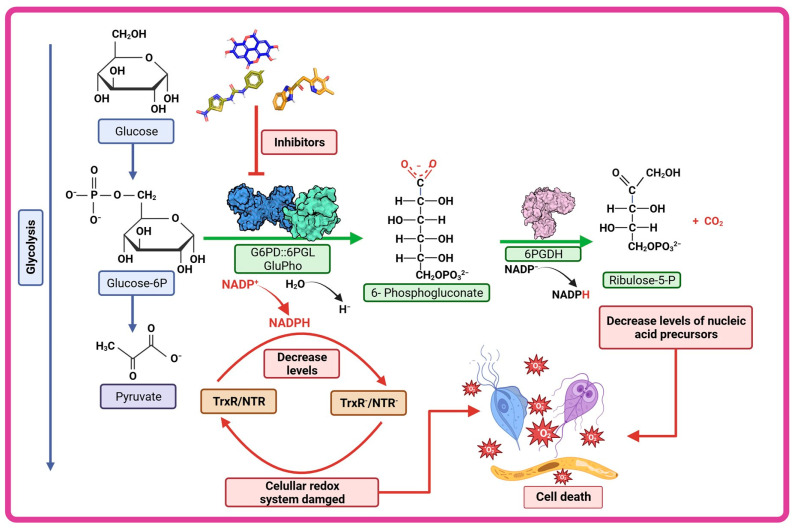
Therapeutic strategy of G6PD::6PGL inhibition. The compounds characterized as inhibitors of the fused G6PD enzymes of *G. lamblia*, *T. vaginalis,* and *P. falciparum* (GlG6PD::6PGL, TvG6PD::6PGL and GluPho respectively) could be altering the cellular redox system of the parasites by decreasing NADPH levels and consequently increasing reactive oxygen species and could also decrease nucleotide precursors such as ribose-5-phosphate and thus lead to the death of the parasite.

**Table 1 microorganisms-12-00112-t001:** Kinetic parameters for both G6PD and 6PGL domains in the fused enzymes PfGluPho, GlG6PD::6PGL, and TvG6PD::6PGL, along with the individually cloned G6PD domain. Kinetic parameters for the individually cloned 6PGL domain have been obtained.

Organism	Enzyme	Kinetic Parameters				Reference
		K_m_ G6P (µM)	K_m_ NADP^+^ (µM)	Km 6PGL (µM)	k_cat_ (s^−1^)	
*P. falciparum*	G6PD	33.2	6.1	-	Nd	[104]
	6PGL	-	-	172	106	[104]
	PfGluPho	19.2	6.5	-	8.6	[104]
*G. lamblia*	G6PD	94.2	26.7	-	0.05	[87]
	6PGL	-	-	51.5	31.8	[87]
	G6PD::6PGL	18.1	13	-	31.8	[96]
*T. vaginalis*	G6PD	Nd	Nd	Nd	Nd	-
	6PGL	Nd	Nd	Nd	Nd	-
	G6PD::6PGL	210	27	-	147	[84]
*Homo sapiens*	G6PD	38.5	6.2	-	233	[105]
	6PGL	-	-	242	505215	[104]

Nd: Not determined.

**Table 2 microorganisms-12-00112-t002:** Reported inhibitor values for the G6PD::6PGL enzymes of the parasites: *P. falciparum*, *P. vivax*, *G. lamblia*, *T. vaginalis*.

Enzyme	Inhibitor	IC_50_ (µM)	*k*_2_ (M^−1^S^−1^)	Type Inhibition	Culture IC_50_ (µM)	Reference
PfGluPho	C276-1187	127	ND	G6P: NC; NADP^+^ : NC	ND	[37,104]
	CB22	4.6	ND	G6P: NC; NADP^+^ : NC	ND	[107]
	CB61	5.1	ND	G6P: MT; NADP^+^ : NC	ND	[107]
	CB62	11.6	ND	ND	ND	[107]
	CB63	1.7	ND	G6P: M; NADP^+^ : NC	ND	[107]
	CB64	19.9	ND	ND	ND	[107]
	CB70	4.5	ND	G6P: MT; NADP^+^ : NC	ND	[107]
	CB103	1.1	ND	G6P: MT; NADP^+^ : NC	ND	[107]
	CB104	7.6	ND	G6P: MT; NADP^+^ : NC	ND	[108]
	ML276	0.889	ND	ND	2.6	[108]
	ML304	0.190	ND	G6P: C; NADP^+^ : MT	0.190	[111]
	Ellagic acid	0.076	ND	G6P: MT; NADP^+^ : MT	0.105	[110]
	21 (vz0527)	1.72	ND	ND	ND	[109]
	4 (vz1732)	0.9	ND	G6P: NC; NADP^+^ : NC	ND	[109]
	SBI-0797750	0.0067	ND	G6P: C; NADP^+^ : MT	0.073	[112]
PvGluPho	Ellagic acid	0.032	ND	G6P: MT; NADP^+^ : MT	ND	[109]
	ML304	15.3	ND	G6P: C; NADP^+^ : MT	ND	[109]
	21 (vz0527)	0.2	ND	ND	ND	[109]
	4 (vz1732)	0.2	ND	ND	ND	[109]
	SBI-0797750	31	ND	G6P: MT; NADP^+^ : MT	ND	[112]
GlG6PD::6PGL	H-BZM-2	37	2.3	ND	36	[113]
	O_2_N-BZM-7	12	1.92	ND	14	[113]
	O_2_N-BZM-9	20	6.13	ND	17	[113]
	CNZ-7	150	0.97	ND	8.7	[114]
	CNZ-8	80	0.60	ND	15	[114]
	CMC-1	70	1.88	ND	15	[114]
	FLP-2	256	0.82	ND	24.1	[114]
TvG6PD::6PGL	O_2_N-BZM-7	22	0.8	ND	6	[113]
	O_2_N-BZM-9	240	1.6	ND	4	[113]
	CNZ-3	93	0.66	G6P: NC; NADP^+^ : C	ND	[115]
	CNZ-17	356	0.38	G6P: UC; NADP^+^ : UC	ND	[115]
	JMM-3	155	0.33	G6P: UC; NADP^+^ : UC	ND	[115]
	MCC-7	260	0.26	G6P: NC; NADP^+^ : UC	ND	[115]

ND: Non determinate. C: Competitive-type inhibition. MT: Mixed-type inhibition. NC: Noncompetitive-type inhibition. UC: Uncompetitive-type inhibition.

## Data Availability

Not applicable.

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
