# Peer review of "Fused Enzyme Glucose-6-Phosphate Dehydrogenase::6-Phosphogluconolactonase (G6PD::6PGL) as a Potential Drug Target in Giardia lamblia, Trichomonas vaginalis, and Plasmodium falciparum"

_microorganisms, 2024, doi:10.3390/microorganisms12010112_

Round 1
Reviewer 1 Report
Comments and Suggestions for Authors
A major public health problem is the limited availability of drugs for the treatment of parasitic diseases and the emergence of drug-resistant strains. Indeed, diseases such as malaria, giardiasis and trichomoniasis require effective treatment to prevent their spread and continue treatment in patients. In this regard, it is important to look for new treatment methods that would be more effective, specific and have less impact. One such approach is rational drug design. This is a technique that allows imaging researchers to both study the specific targets needed for parasites to reproduce and create molecules that can effectively inhibit them. Rational design involves computer modeling and structure-function analysis of parasite protein targets. This allows researchers to identify key regions of the protein that interact with existing drug molecules and create drugs that can effectively bind to the sites in the body, allowing the protein to function normally and thus causing parasites to spread. Rational drug design can be particularly useful in developing new drugs to combat drug-resistant strains of parasites. Based on information about the detection and function of protein targets that are identified antiparasitic drugs, researchers can predict the effectiveness of various drug compounds and select the most promising candidates for further testing. Therefore, rational drug design is important in modern medicine, which helps researchers to develop new drugs to treat parasitic diseases with higher efficacy and increased side effects. This approach could be used to combat the growing problem of parasite control with existing drugs and ensure future treatments are effective.
It is very important to find new solutions for the treatment of protozoal diseases that are effective, safe and accessible to patients. Indeed, side effects from drugs can be unpleasant and even dangerous, so it is important to develop medications that have a minimum of such effects. In addition, the availability of drugs is also an important aspect. The global problem of protozoal diseases is often associated with low incomes and poor access to treatment in some regions. Therefore, the development of drugs that will be affordable for a wide audience is a priority.
The work can be accepted.
Reviewer 2 Report
Comments and Suggestions for Authors
This is a review article.
The title should be changed, as not all the protozoa that affect humans are covered.
The revision should be directed towards a globalised world and not towards certain regions, continents or locations. E.g. line 51, it's not just a Latin American problem.
In general, the article is well written and covers some protozoa. It should be expanded to include some of great importance such as Leishmania, Isospora, Entamoeba... and perhaps helminthosis That would live up to the title. So I hope the authors will do that.
Round 2
Reviewer 2 Report
Comments and Suggestions for Authors
The authors changed the title to suit the parasites commented on: Fused enzyme Glucose-6-phosphate dehydrogenase::6-phos- 2 phogluconolactonase (G6PD::6PGL) as a potential drug target 3 in parasites of medical importance.
Some of the points commented on have also changed.
They explained the application of some protozoa and the non-inclusion of others. I think this was enlightening for its acceptance.